# Aneurysmal Subarachnoid Hemorrhage in Hospitalized Patients on Anticoagulants—A Two Center Matched Case-Control Study

**DOI:** 10.3390/jcm12041476

**Published:** 2023-02-13

**Authors:** Michael Veldeman, Tobias Rossmann, Miriam Weiss, Catharina Conzen-Dilger, Miikka Korja, Anke Hoellig, Jyri J. Virta, Jarno Satopää, Teemu Luostarinen, Hans Clusmann, Mika Niemelä, Rahul Raj

**Affiliations:** 1Department of Neurosurgery, University of Helsinki and Helsinki University Hospital, 00260 Helsinki, Finland; 2Department of Neurosurgery, RWTH Aachen University Hospital, 52074 Aachen, Germany; 3Department of Neurosurgery, Neuromed Campus, Kepler University Hospital, 4021 Linz, Austria; 4Department of Neurosurgery, Kantonsspital Aarau, 5001 Aarau, Switzerland; 5Division of Anesthesiology, Department of Anesthesiology, Intensive Care and Pain Medicine, University of Helsinki and Helsinki University Hospital, 00260 Helsinki, Finland; 6Anaesthesiology and Intensive Care, University of Helsinki and Helsinki University Hospital, 00260 Helsinki, Finland

**Keywords:** subarachnoid hemorrhage, intracranial aneurysm, direct oral anticoagulants, vitamin K antagonists

## Abstract

**Objective**—Direct oral anticoagulants (DOAC) are replacing vitamin K antagonists (VKA) for the prevention of ischemic stroke and venous thromboembolism. We set out to assess the effect of prior treatment with DOAC and VKA in patients with aneurysmal subarachnoid hemorrhage (SAH). **Methods**—Consecutive SAH patients treated at two (Aachen, Germany and Helsinki, Finland) university hospitals were considered for inclusion. To assess the association between anticoagulant treatments on SAH severity measure by modified Fisher grading (mFisher) and outcome as measured by the Glasgow outcome scale (GOS, 6 months), DOAC- and VKA-treated patients were compared against age- and sex-matched SAH controls without anticoagulants. **Results**—During the inclusion timeframes, 964 SAH patients were treated in both centers. At the time point of aneurysm rupture, nine patients (0.93%) were on DOAC treatment, and 15 (1.6%) patients were on VKA. These were matched to 34 and 55 SAH age- and sex-matched controls, re-spectively. Overall, 55.6% of DOAC-treated patients suffered poor-grade (WFNS_4–5_) SAH compared to 38.2% among their respective controls (*p* = 0.35); 53.3% of patients on VKA suffered poor-grade SAH compared to 36.4% in their respective controls (*p* = 0.23). Neither treatment with DOAC (aOR 2.70, 95%CI 0.30 to 24.23; *p* = 0.38), nor VKA (aOR 2.78, 95%CI 0.63 to 12.23; *p* = 0.18) were inde-pendently associated with unfavorable outcome (GOS_1–3_) after 12 months. **Conclusions**—Iatrogenic coagulopathy caused by DOAC or VKA was not associated with more severe radiological or clinical subarachnoid hemorrhage or worse clinical outcome in hospitalized SAH patients.

## 1. Introduction

Direct oral anticoagulants (DOAC) are progressively replacing vitamin K antagonists (VKA) for the prevention of ischemic stroke and venous thromboembolism [1,2,3,4]. Their use eliminates the need for constant monitoring of coagulation status due to more predictable pharmacokinetics and fewer drug interactions compared to VKA [5].

In contrast to VKA’s indirect anticoagulant effect, DOAC directly inhibits a single clotting enzyme, factor-Xa, in case of apixaban, edoxaban and rivaroxaban, and thrombin in case of dabigatran [6]. Their relatively short half-life (around 12 h) allows spontaneous reversal of effects when elective or subacute surgery is warranted [7,8]. Idarucizumab, a costly monoclonal antibody fragment, reverses the anticoagulant effects of dabigatran only [9]. Andexanet alfa was approved as a reversal agent for rivaroxaban and apixaban in 2018, and other candidate antidotes currently undergoing clinical testing will follow [10,11].

Patients carrying an intracranial aneurysm are at risk of aneurysm rupture causing subarachnoid hemorrhage (SAH) [12]. Depending on the morphology and orientation of the aneurysm, bleeding can extend into brain parenchyma or ventricular system, causing ICH or intraventricular hemorrhage (IVH). The resulting acute increase of intracranial pressure (ICP) induces a drop in cerebral perfusion and can cause transient intracranial circulatory arrest [13] or sudden death [14]. Bleeding ceases once a clot effectively plugs the rupture site, a process requiring platelet adherences alongside functioning coagulation for thrombus stabilization [15]. Survivors need occlusion of the offending aneurysm from its parent vessel either by endovascular or surgical means, to prevent rebleeding [16]. The risk of re-rupture is the highest within the first hours [17]. Waiting to treat until effects are reversed by drug metabolization, is not desirable in SAH patients.

In two case-control analyses, VKA was associated with an increased risk of SAH [18,19]. The effects of DOAC intake in comparison with VKA on the severity, course, and outcome of SAH have not been investigated. Iatrogenic coagulopathy is suspected to worsen clinical and radiological severity of bleeding and increase the risk profile of aneurysm occlusion procedures as well as treatment of associated acute hydrocephalus. We set out to assess the bleeding severity and outcome after SAH, in patients on VKA and DOAC, in a two-center observational cohort study.

## 2. Methods

### 2.1. Patient Population and Study Design

All consecutive SAH patients who were treated at two university hospitals between 2010 and 2019 (RWTH Aachen University Hospital, Aachen, Germany) and between 2014 and 2019 (Helsinki University Hospital, Helsinki, Finland) were considered for inclusion. Patients aged 18 years or older with confirmation of ruptured aneurysm in either CT- or conventional cerebral angiography were included. From 2014 onward, the prospective data collection in Aachen was part of a previously registered observational study (NCT02142166) and was approved by the local ethics committee of the Medical Faculty of RWTH Aachen University (EK 062/14). The Helsinki database was retrospectively collected (HUS/125/2018). Due to the retrospective data collection of the remainder of data, the need for patient consent was waived by both local institutional research committees. Relevant patient baseline and SAH-specific data were extracted from existing electronic health records. The intake of antithrombotic drugs at time of SAH, in form of either antiplatelet drugs, VKA or DOAC, was noted. Coagulation status on admission consisting of platelet count and prothrombin time (PT) expressed as International Normalized Ratio (INR) was extracted. Clinical state on admission was assigned as best GCS performance in 24 h and graded by means of the World Federation of Neurological Surgeons (WFNS) grading scale. Clinical severity was dichotomized into good-grade (WFNS_1–3_) and poor-grade (WFNS_4–5_) SAH.

### 2.2. Standard of Care

In both inclusion centers, occlusion of the offending aneurysm was aimed for within 48 h via either surgical clipping or endovascular occlusion (coiling, flow-diverter stenting, or WEB-device placement). All patients were observed in a dedicated neurointensive care unit. Anticoagulant effects of VKA were acutely reversed by application of a body weight- adjusted dose of PCC until reaching a minimum INR of 1.2 [20]. Before antidotes were available, DOAC-treated patients also received PCC, but since 2014, the effect of dabigatran was reversed with idarucizumab [21,22]. In case of acute hydrocephalus, an external ventricular drain was placed prior to aneurysm occlusion but after anticoagulant reversal. After aneurysm occlusion, all patients received a wake-up test after which neurological assessability was continuously strived for. All patients were prophylactically treated with oral or intravenous nimodipine. More elaborate institutional treatment algorithms have been published previously [23,24].

### 2.3. Design and Outcome Parameters

To assess the effect of prior DOAC treatment on SAH severity and outcome, DOAC-treated patients were age- and sex-matched to non-DOAC controls with the aim of correcting for confounding comorbidities associated with anticoagulant use. Patients using VKA or DOAC and patients not using any antithrombotics were matched based on age and sex.

The same procedure was repeated for patients on VKA. Controls were only selected from patients from the same institution which were neither on VKA/DOAC nor any other antithrombotic treatment. Primary outcome was defined as radiological hemorrhage severity as measured by the modified Fisher scale [25] with additional assessment of the presence of ICH. Secondary outcome was defined as clinical hemorrhage severity (WFNS) [26] in form of incidence of poor-grade (WFNS_4–5_) SAH, occurrence of delayed cerebral ischemia (DCI), in-hospital mortality, and clinical outcome after 12 months as measured via the Glasgow outcome scale (GOS) [27]. The GOS was dichotomized into favorable (GOS_4–5_) and unfavorable outcome (GOS_1–3_). The standing definition of clinical DCI by Vergouwen et al. [28] was applied whenever possible, but for the unconscious patient, diagnosis was based on perfusion CT-imaging or multimodal neuromonitoring [23,24,29].

### 2.4. Statistical Analysis

Numerical data are presented as median and interquartile ranges (Q_1_–Q_3_) due to small subgroup sizes. Categorical variables are provided as absolute case numbers and percentages. For nominal data, either the *χ*^2^-Test or Fisher’s exact test was used as appropriate based on group size. For continuous data, the Mann–Whitney *U* Test was used. A 3–4:1 age- and sex-matching of cases to controls was performed using the “ccmatch” function in STATA 17 (StataCorp LCC, College Station, TX, USA) selecting controls for each case only from patients coming from the same institution of treatment. Factors associated with occurrence of unfavorable outcome were assessed in a logistic regression model. Explanatory variables were included based on univariate results presenting with *p*-value < 0.10 or based on clinical relevance. Variables were tested for outliers via plotting and multicollinearity was evaluated via the assessment of the Variance Inflation Factor with a cut-off of 2.5. Multivariable-adjusted odds ratios (aORs) for SAH were estimated by conditional logistic regression. The latter statistical analyses were performed using IBM SPSS Statistics 25 (IBM Inc., Chicago, IL, USA). Statistical significance was defined as a two-sided *p* < 0.05.

## 3. Results

### 3.1. Patient Inclusion and Baseline Characteristics

During the inclusion timeframes, 964 SAH patient were treated, of whom 595 were in Helsinki and 369 were in Aachen. Sixty-six patients with an initial moribund presentation, none of which was treated with anticoagulants, were excluded. Moribund presentation was defined as clinical brainstem herniation based on bilateral fixed pupils or global supratentorial ischemic injury on imaging. A total of nine patients (0.93%) received DOAC treatment before suffering SAH, of whom two patients were treated with apixaban, a single patient with dabigatran, and six patients with rivaroxaban. No patients on edoxaban were identified. Seven patients received DOAC for non-valvular atrial fibrillation and two patients due to a history of deep vein thrombosis. No patients in the DOAC-control group suffered from type 2 diabetes.

A total of 15 (1.6%) SAH patients on VKA at the time point of aneurysm rupture were identified. Thereof, three patients were on phenprocoumon and 12 patients were on warfarin, prescribed for atrial fibrillation in 13 patients, and in three patients for a history of pulmonary embolism. No patients anticoagulated for mechanical heart valves were identified. As expected, patients on VKA presented with higher initial INR (2.2 (1.4–3.0) vs. 1.1 (1.0–1.1); *p* < 0.001). From the remainder of patients, 34 age- and sex-matched controls for DOAC cases and 55 controls for VKA cases were identified. An inclusion flow-chart is presented as Figure 1.

The location of the offending aneurysm was similarly distributed between both groups. Aneurysm distribution and size were comparable between VKA cases and controls. A comparison of patient baseline and SAH characteristics is presented in Table 1 and Table 2.

### 3.2. Hemorrhage Severity

Radiological severity (modified Fisher scale) of SAH in patients on oral anticoagulants was similar compared to age- and sex-matched controls. Rates of IVH or ICH were not affected by intake of either DOAC or VKA (see Table 1 and Table 2). A total of 55.6% of DOAC-treated patients suffered poor-grade SAH compared to 38.2% in the control group (*p* = 0.349). In SAH patients on VKA, 53.3% suffered poor-grade SAH vs. 36.4% in age- and sex-matched controls (*p* = 0.234). No patients with aneurysm rebleeding were identified.

### 3.3. Effects on Clinical Outcome

All patients on DOAC required mechanical ventilation (not including anesthesia for aneurysm occlusion) during their hospital stay compared to only 19 (55.9%) patients of matched controls (*p* = 0.014). The rate of patients suffering from sepsis, according to standing criteria, diagnosed by the presence of infection together with systemic manifestations of infection [30], was higher in patients who had received DOAC-treatment (n = 2 (22.2%) vs. n = 1 (2.9%); *p* = 0.043). No differences in the need for mechanical ventilation or the development of sepsis were noted between patients previously on VKA and matched controls. Six patients (66.7%) in the DOAC group compared to 19 (55.9%) controls were classified as unfavorable outcome after 12 months (*p* = 0.560). In VKA-treated patients, 11 (73.3%) patients were classified as unfavorable outcome compared to 26 (47.3%) patients in the matched control group (*p* = 0.073). An overview of clinical outcome parameters is presented in Table 3. Based on univariate logistic regression result (Appendix A), the association between previous DOAC treatment and unfavorable outcome (GOS_1–3_) after 12 months was adjusted for WFNS grading, DCI occurrence, need for mechanical ventilation, and occurrence of sepsis. Introducing these explanatory variables into a conditional multivariable regression resulted in a significant model (χ^2^(5) = 13.837, *p* = 0.017) explaining 36.7% (Nagelkerke *R*^2^) of the variance in occurrence of unfavorable outcome and correctly classifying 74.4% of cases. Prior DOAC use was not independently associated with unfavorable outcome (aOR 2.696, 95%CI 0.300 to 24.228; *p* = 0.376).

A second model was built to assess the association between VKA use and unfavorable outcome resulting in a significant model (χ^2^(5) = 26.364, *p* < 0.001; Nagelkerke *R*^2^ = 0.419) classifying 87.8% of cases correctly. Treatment with VKA had no effect on the occurrence of unfavorable outcome after 12 months (aOR 2.780, 95%CI 0.631 to 12.234; *p* = 0.176) (See Appendix A). Comparing the proportions of unfavorable outcome between DOAC and VKA patients yielded a χ^2^(2) statistic of 4.741 corresponding to a *p*-value of 0.093.

## 4. Discussion

This matched case-control study examined the effect of anticoagulant treatment on severity and outcome of SAH. The prevalence of SAH in patients on anticoagulants is low and constituted around two and a half percent of all SAH patients congruous with population-wide prevalence of anticoagulant intake. Prior intake of DOAC or VKA was not associated with more severe SAH, or a higher risk of unfavorable outcome. Although we have investigated an overall large cohort of SAH patients, the number of patients on anticoagulation is low, which might distort statistical analyses. Additionally, it is noteworthy that we assessed a cohort of only hospitalized patients and cannot exclude the possibility of a higher out-of-hospital death rate for SAH patients using DOAC or VKA.

A higher proportion of patients on DOAC required mechanical ventilation and experienced systemic infections. The main aim of the matching procedure was to correct for comorbidities associated with anticoagulant use and higher age. Patients with an indication for continuous oral anticoagulant based on vascular- or heart disease tend to be older and have higher rates of cardiovascular risk factors. Instead of correcting for each individual factor (which constitutes a non-exhaustive list), we only corrected for age. This could mean there is still a higher degree of “baseline comorbidity” in the coagulant groups, which is not corrected for. This could explain an increased propensity towards infections in patients on prior anticoagulant treatment. However, it does not explain why this effect was only apparent in DOAC patients.

Hypercoagulability plays a role in the pathogenesis of DCI and microthrombus formation has been observed in in vivo animal SAH models [31,32]. Moreover, there exists observational efficacy of heparin anticoagulation in clinical use to prevent DCI [33]. The results of the Aneurysmal Subarachnoid Hemorrhage Trial RandOmizing Heparin (ASTROH)-trial are highly anticipated [34]. An additional residual dysfunctional coagulation in the subacute phase could prove protective against DCI by mitigating microthrombosis formation. In our comparison to matched controls, the rate of DCI was similar in patients on prior anticoagulant treatment. The subgroups of anticoagulated patients proved too small to focus analyses on DCI here. However, as our understanding of dysfunctional coagulation microvascular hypercoagulability improves, the potentially beneficial side effects of iatrogenic anticoagulation may become clearer.

Approximately two percent of the general population in Western countries is currently on VKA [35]. Mainly driven by population aging, the use of anticoagulants for prevention of stroke and venous thromboembolisms will only continue rising. Unfortunately, this has been paralleled by an increase in anticoagulant-related ICH [36]. As evidence of safety and effectiveness accumulates, new oral anticoagulants will probably supersede VKA for most indications [37]. Although ICH has not been an endpoint in randomized trials comparing DOAC with warfarin [1,2,3,4], observational data suggests DOAC-associated ICH presents with smaller bleeding volumes and lower hematoma expansion rates [38,39,40]. Along with a shorter half-life, the targeting of a single clotting factor by DOAC—opposed to all vitamin K-dependent factors—allows coagulation to recover more swiftly. The successive development of antidotes will further improve the risk profile of anticoagulants in favor of DOAC over VKA.

Increasing accessibility of magnetic resonance imaging has resulted in a higher incidental detection of aneurysms [41,42,43]. It remains unclear to what extent anticoagulants might be harmful in those patients. Data on the risk of antithrombotics and SAH has been conflicting. Garbe et al. identified phenprocoumon, clopidogrel/ticlopidine, and acetylsalicylic acid intake, to be associated with increased risk of SAH [18]. In contrast, Risselada et al. demonstrated a similar result for VKA but not for platelet aggregation inhibitors [19]. Multiple smaller observational studies refute that the risk of aneurysm rupture is increased in patients receiving systemic anticoagulation [44,45,46]. Acetylsalicylic acid is currently being tested in the ongoing PROTECT-U trial (Prospective Randomized Open-label Trial to Evaluate risk faCTor management in patients with Unruptured intracranial aneurysms), to address aneurysm wall inflammation, but might be a hazard to the consequences of its anti-aggregating affect [47].

Nonetheless, if rupture occurs, thrombus formation at the rupture site may take more time, presumably leading to more severe bleeding. Our series demonstrates how hospitalized patients on anticoagulants presenting with SAH are not more severely affected than patients without anticoagulant drugs. The opposite effect was demonstrated in a Dutch series of 15 SAH patients on anticoagulant of which 14 were either dead or highly dependent after three months follow-up [48]. The author’s explanation for this effect is that the worse clinical condition patients are in from the outset. It is possible that a faster emergency service’s response time in densely populated areas (such as the Netherlands) could increase the number of patients reaching medical services alive, but in a worse clinical state.

Our understanding of aneurysm rupture risk is partially based on patients presenting with SAH [49]. This means patients who suffered SAH-induced sudden death are not represented in these statistics. Likewise, the safety of DOAC and anticoagulants in patients with incidental finding of unruptured intracranial aneurysms cannot be resolved solely by looking at SAH patients since anticoagulated patients suffering SAH-related sudden death are not represented. Nevertheless, even in large-scale atrial fibrillation registries of patients on anticoagulants, autopsy-confirmed cause of death—such as aneurysm rupture—has not been a feasible endpoint as autopsy is not consistently performed [50].

## 5. Limitations

Besides this selection bias, further limitations of this analysis are the retrospective design and the use of statistically constructed control groups. Sample sizes remain small and statistical results must be interpreted with caution. Although our case-matching algorithm corrects for confounding differences in baseline comorbidities associated with patient age and gender, information and potential detection bias cannot be corrected for. Blood concentrations of DOAC, which vary widely between peak and trough levels, were not routinely available. With the availability and increasing affordability of antidotes, the safety issue of anticoagulants will become less relevant when considering risk of hemorrhagic complications after emergent invasive procedures. However, initial hemorrhage severity and the extent of early brain injury it causes is—despite reversals of anticoagulant effect upon hospital submission—responsible for initiating delayed cerebral ischemia. Therefore, studying anticoagulant use in patients suffering subarachnoid hemorrhage will remain relevant.

## 6. Conclusions

In this two-center observational study of hospitalized aneurysmal SAH patients, iatrogenic coagulopathy, either caused by DOAC or VKA, was not associated with more severe radiological or clinical subarachnoid hemorrhage compared to matched controls. Intake of anticoagulants was not associated with a higher risk of unfavorable outcome. The rate of unfavorable outcome was similar between patients on VKA compared to DOAC. Nevertheless, the safety issue of DOAC and anticoagulants in patients with unruptured intracranial aneurysms cannot be resolved based on this study as out-of-hospital deaths were not accounted for.

## Figures and Tables

**Figure 1 jcm-12-01476-f001:**
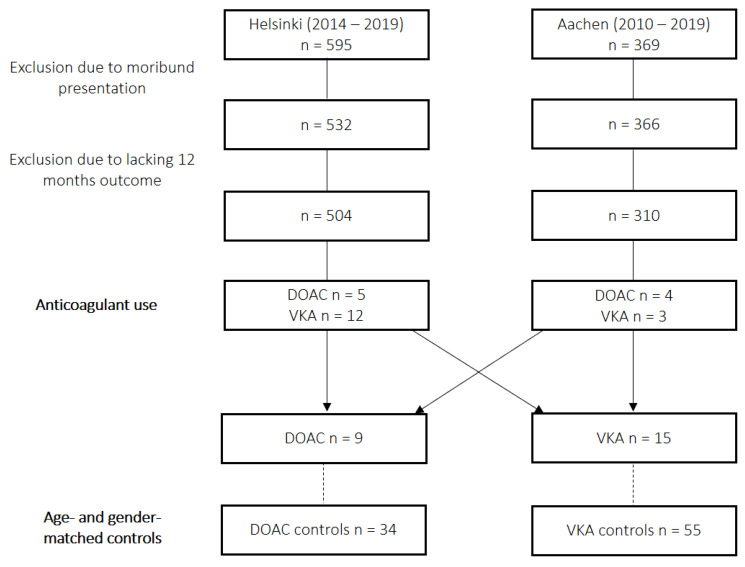
Inclusion flow-chart. DOAC, direct oral anticoagulant; VKA, vitamin K antagonist.

**Table 1 jcm-12-01476-t001:** Comparison of patients on direct oral anticoagulants (DOAC) with age- and sex-matched controls without antithrombotic treatment.

	All (n = 43)	DOAC (n = 9)	Controls (n = 34)	*p*-Value
**Demographics**				
Age—yrs.—median (Q_1_–Q_3_)	67.5 (62.1–76.5)	66.6 (61.5–78.7)	68.1 (61.9–75.6)	0.895
Sex—Female/Male- no. (%)	25 (58.1)/18 (41.9)	5 (55.6)/4 (44.4)	20 (58.8)/14 (41.2)	0.860
**Comorbidity—no. (%)**				
Hypertension	26 (60.5)	5 (55.6)	21 (61.8)	0.735
Smoking	13 (30.2)	2 (22.2)	11 (32.4)	0.556
Diabetes type 2	2 (4.7)	2 (22.2)	0	n/a
Coronary artery disease	2 (4.7)	1 (11.1)	1 (2.9)	0.301
**Coagulation status on admission**				
INR—median (Q_1_–Q_3_)	1.1 (1.0–1.1)	1.1 (1.1–1.2)	1.0 (1.0–1.1)	0.081
Thrombocyte count (10^3^/µL)—median (Q_1_–Q_3_)	241.0 (209.0–291.0)	235.0 (185.0–452.0)	243.5 (211.5–283.5)	0.895
**Aneurysm location—no. (%)**				
Acomm	16 (37.2)	4 (44.4)	12 (35.3)	0.256
MCA	10 (23.3)	1 (11.1)	9 (26.5)	
ICA (incl. Pcomm)	11 (25.6)	4 (44.4)	7 (20.6)	
Others	6 (14.0)	0 (0)	6 (17.6)	
Ant. circulation	37 (86.0)	9 (100.0)	28 (82.4)	0.174
Max. diameter (mm)—median (Q_1_–Q_3_)	6.0 (4.3–9.0)	5.0 (3.0–7.0)	7.0 (4.9–9.5)	0.135
**Aneurysm occlusion—no. (%)**				
Clipping/Endovascular	13 (30.2)/30 (69.8)	1 (11.1)/8 (88.9)	12 (35.3)/22 (64.7)	0.160
**Hemorrhage severity**				
**WFNS grade—no. (%)**				0.777
Grade 1	9 (20.9)	1 (11.1)	8 (23.5)	
Grade 2	10 (23.3)	2 (22.2)	8 (23.5)	
Grade 3	6 (14.0)	1 (11.1)	5 (14.7)	
Grade 4	6 (14.0)	1 (11.1)	5 (14.7)	
Grade 5	12 (27.9)	4 (44.4)	8 (23.5)	
Poor-grade SAH (WFNS_4–5_)	18 (41.9)	5 (55.6)	13 (38.2)	0.349
**Modified Fisher scale—no. (%)**				0.558
Grade 1	8 (18.6)	1 (11.1)	7 (20.6)	
Grade 2	5 (11.6)	2 (22.2)	3 (8.8)	
Grade 3	9 (20.9)	1 (11.1)	8 (23.5)	
Grade 4	21 (48.8)	5 (55.6)	16 (47.1)	
IVH	26 (60.5)	7 (77.8)	19 (55.9)	0.296
ICH	15 (34.9)	4 (44.4)	11 (32.4)	0.499
Acute hydrocephalus	27 (62.8)	8 (88.9)	19 (55.9)	0.069

Acomm, aneurysm of the anterior communicating artery; ICH, intracerebral hemorrhage; INR, international normalized ratio; IVH, intraventricular hemorrhage; MCA, aneurysm of the middle cerebral artery; Pcomm, aneurysm of the posterior communicating artery; Q_1_, first quartile; Q_3_, third quartile; WFNS, World Federation of Neurological Surgeons.

**Table 2 jcm-12-01476-t002:** Comparison of patients on vitamin K antagonists (VKA) with age- and sex-matched controls without antithrombotic treatment.

	All (n = 70)	VKA (n = 15)	Controls (n = 55)	*p*-Value
**Demographics**				
Age—yrs.—median (Q_1_–Q_3_)	73.0 (65.0–76.8)	73.0 (66.4–76.6)	73.0 (63.8 - 77.6)	0.869
Sex—Female/Male- no. (%)	38 (54.3)/32 (45.7)	8 (53.3)/7 (46.7)	30 (54.5)/25 (45.5)	0.933
**Comorbidity—no. (%)**				
Hypertension	39 (55.7)	13 (86.7)	26 (47.3)	0.006
Smoking	17 (24.3)	3 (20.0)	14 (25.5)	0.662
Diabetes type 2	6 (8.6)	3 (20.0)	3 (5.5)	0.074
Coronary artery disease	3 (4.3)	3 (20.0)	0 (0)	**0.001**
**Coagulation status on admission**				
INR—median (Q_1_–Q_3_)	1.1 (1.0–1.2)	2.2 (1.4–3.0)	1.1 (1.0–1.1)	**<0.001**
Thrombocyte count (10^3^/µL)—median (Q_1_–Q_3_)	241.0 (194.5–278.0)	221.0 (171.0–260.0)	242.0 (208.3–290.5)	0.169
**Aneurysm location—no. (%)**				
Acomm	23 (32.9)	5 (33.3)	18 (32.7)	0.094
MCA	16 (22.9)	2 (13.3)	14 (25.5)	
ICA (incl. Pcomm)	14 (20.0)	1 (6.7)	13 (23.6)	
Others	17 (24.3)	7 (46.7)	10 (18.2)	
Ant. circulation	53 (75.7)	8 (53.3)	45 (81.8)	0.023
Max. diameter (mm)—median (Q_1_–Q_3_)	6.0 (4.0–8.0)	6.2 (4.5–9.0)	6.0 (4.0–8.0)	0.873
**Aneurysm occlusion—no. (%)**				
Clipping/Endovascular	23 (32.9)/47 (67.1)	3 (20.0)/12 (80.0)	20 (36.4)/35 (63.6)	0.232
**Hemorrhage severity**				
**WFNS grade—no. (%)**				0.331
Grade 1	23 (32.9)	6 (40.0)	17 (30.9)	
Grade 2	9 (12.9)	1 (6.7)	8 (14.5)	
Grade 3	10 (14.3)	0 (0)	10 (18.2)	
Grade 4	12 (17.1)	3 (20.0)	9 (16.4)	
Grade 5	16 (22.9)	5 (33.3)	11 (20.0)	
Poor-grade SAH (WFNS_4–5_)	28 (40.0)	8 (53.3)	20 (36.4)	0.234
**Modified Fisher scale—no. (%)**				0.528
Grade 1	11 (15.7)	2 (13.3)	9 (16.4)	
Grade 2	5 (7.1)	2 (13.3)	3 (5.5)	
Grade 3	17 (24.3)	2 (13.3)	15 (23.3)	
Grade 4	37 (52.9)	9 (60.0)	28 (50.9)	
IVH	40 (57.1)	11 (73.3)	29 (52.7)	0.153
ICH	22 (31.4)	5 (33.3)	17 (30.9)	0.858
Acute hydrocephalus	46 (65.7)	12 (80.0)	34 (61.8)	0.189

Acomm, aneurysm of the anterior communicating artery; ICH, intracerebral hemorrhage; INR, international normalized ratio; IVH, intraventricular hemorrhage; MCA, aneurysm of the middle cerebral artery; Pcomm, aneurysm of the posterior communicating artery; Q_1_, first quartile; Q_3_, third quartile; WFNS, World Federation of Neurological Surgeons. Significant *p*-values (<0.05) are marked in bold.

**Table 3 jcm-12-01476-t003:** Outcome comparison of patients on either direct oral anticoagulants (DOAC) or vitamin K antagonists (VKA) with age- and sex-matched controls without antithrombotic treatment.

	All (n = 43)	DOAC (n = 9)	Controls (n = 34)	*p*-Value
**ICU-related complications**				
Mechanical ventilation—no. (%)	28 (65.1)	9 (100)	19 (55.9)	0.014
Sepsis—no. (%)	3 (7.0)	2 (22.2)	1 (2.9)	0.043
DCI occurrence—no. (%)	15 (34.9%)	2 (22.2)	13 (38.2)	0.370
**Clinical outcome**				
In-hospital mortality—no. (%)	7 (16.3)	3 (33.3)	4 (11.8)	0.119
GOS 12 months—no. (%)				0.892
Good recovery	10 (23.3)	2 (22.2)	8 (23.5)	
Moderate disability	8 (18.6)	1 (11.1)	7 (20.6)	
Severe disability	14 (32.6)	3 (33.3)	11 (32.4)	
Vegetative state	1 (2.3)	0 (0)	1 (2.9)	
Dead	10 (23.3)	3 (33.3)	7 (20.6)	
Unfavorable outcome (GOS_1–3_)	25 (58.1)	6 (66.7)	19 (55.9)	0.560
	**All (n = 70)**	**VKA (n = 15)**	**Controls (n = 55)**	***p*-Value**
**ICU-related complications**				
Mechanical ventilation—no. (%)	50 (71.4)	38 (69.1)	12 (80.0)	0.407
Sepsis—no. (%)	9 (12.9)	7 (12.7)	2 (13.3)	0.950
DCI occurrence—no. (%)	21 (30.0)	17 (30.9)	4 (26.7)	0.751
**Clinical outcome**				
In hospital mortality—no. (%)	19 (27.1)	3 (20.0)	16 (29.1)	0.483
GOS 12 months—no. (%)				0.907
Good recovery	22 (31.4)	3 (20.0)	19 (34.5)	
Moderate disability	11 (15.7)	1 (6.7)	10 (18.2)	
Severe disability	13 (18.6)	6 (40.0)	7 (12.7)	
Vegetative state	3 (4.3)	1 (6.7)	2 (3.6)	
Dead	21 (30.0)	4 (26.7)	17 (30.9)	
Unfavorable outcome (GOS_1–3_)	37 (52.9)	11 (73.3)	26 (47.3)	0.073

DCI, delayed cerebral ischemia; DOAC, direct oral anticoagulants; GOS, Glasgow outcome scale; ICU, intensive care unit; VKA, vitamin K antagonists.

## Data Availability

The date on which the analyses are based can be made available upon reasonable request from qualified researchers by contacting the corresponding author.

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
