# Peer review of "Aneurysmal Subarachnoid Hemorrhage in Hospitalized Patients on Anticoagulants—A Two Center Matched Case-Control Study"

_jcm, 2023, doi:10.3390/jcm12041476_

Round 1

Reviewer 1 Report

The authors present their multi-institutional observation of a cohort of patients on DOAC and VKA and compare the subgroups with all the aneurysmal SAH they care for in this population.

Overall a very intersting study with some comments for clarification: 

1. Why do you think a higher proportion of patients on a DOAC experience systemic infection?

2. Why do you think a lower proportion of patients on a DOAC experience DCI but not VKA?

The finding of DOAC/VKA use not being an independent risk factor for poor outcome is interesting but I would have hoped for a little more development in the discussion of the findings in your tables. 

Author Response

Reviewer #1

The authors present their multi-institutional observation of a cohort of patients on DOAC and VKA and compare the subgroups with all the aneurysmal SAH they care for in this population.

Overall a very interesting study with some comments for clarification:

  1. Why do you think a higher proportion of patients on a DOAC experience systemic infection?

We thank the reviewer for this very relevant comment. The main aim of the matching procedure was to correct for comorbidities associated with anticoagulant use and higher age. Patients with an indication for continuous oral anticoagulant based on arterial- venous- or heart disease, tend to be older and have higher rates of hypercholesterolemia, arterial hypertension, type 2 diabetes, nicotine consumption, etc.. Instead of correcting for each individual cardiovascular risk factors (which constitutes a non-exhaustive list), we have only corrected for age (and gender).

This could mean there is still a higher degree of “base line comorbidity” which is not corrected for. This could leave patients taking anticoagulants with a higher propensity towards infection during a long ICU stays in reduced clinical state.

This however does not explain why DOAC patients would be more susceptible to systemic infections than patients on VKA. The indications for both groups of anticoagulants were the same (atrial fibrillation, AF or pulmonary embolism, PE) and were similarly distributed (7/9 with AF and 2/9 with PE in the DOAC group versus 13/15 with AF and 2/15 with PE in the VKA group). No patients with prosthetic heart valves, predisposing to valvar vegetation or septic emboli, were included in neither of both groups. The origin of the difference in susceptibility between patients receiving DOAC or VKA remains unclear. Alternatively, a Type 1 error is plausible due to a sampling effect induced the small group size of DOAC patients.

This interpretations is now incorporated into the Discussion’s section of text as follows:

“A higher proportion of patients on DOAC required mechanical ventilation and experienced systemic infections. The main aim of the matching procedure was to correct for comorbidities associated with anticoagulant use and higher age. Patients with an indication for continuous oral anticoagulant based on vascular- or heart disease, tend to be older and have higher rates of cardiovascular risk factors. Instead of correcting for each individual factors (which constitutes a non-exhaustive list), we only corrected for age. This could mean there is still a higher degree of “base line comorbidity” in the coagulant groups, which is not corrected for. This could explain an increased propensity towards infections in patients priorly on anticoagulants. It remains however unclear why this effect was only apparent in DOAC patients. ”

  1. Why do you think a lower proportion of patients on a DOAC experience DCI but not VKA?

This is also a very valid remark. However, although absolute DCI-rates differed (22.2% in DOAC patients versus 30.9% in VKA patients), no significant differences to matched controls were noted. Nonetheless, during initial study design, DCI and DCI-related infarction were considered as main outcome factors. This under the hypothesis that anticoagulation during the acute phase might mitigate microthrombosis formation and therefore reduce risk of DCI. Microthrombus formation has been observed in in-vivo animal SAH models as a trigger of DCI (1, 2) and there has been efficacy of heparin anticoagulation in clinical use to prevent DCI (3). The results of the Aneurysmal Subarachnoid Hemorrhage Trial RandOmizing Heparin (ASTROH)-trial are further anticipated. Because of slight differences in DCI-treatment between both inclusion centers and the lack of DOAC levels to quantify anticoagulant effect, it made more sense from a clinical point of view in these small collectives of patients (found in 964 SAH patients), to focus on initial clinical and radiological severity. Nonetheless, it could very well be that the acute reversal of anticoagulant effect in VKA patients (via factor substitution) might have had the exact opposite effect and induced microvascular thrombophilia promoting DCI. An effect which is not at play in the DOAC group in which only a single patient was reversed with idarucizumab (all other patients were on DOAC for which no antidote was available at the time). To address this issue, this text has now been added to the discussion:

“Hypercoagulability plays a role in the pathogenesis of DCI and microthrombus formation has been observed in in-vivo animal SAH models. (1, 2) Moreover, there exists observational efficacy of heparin anticoagulation in clinical use to prevent DCI. (3) The results of the Aneurysmal Subarachnoid Hemorrhage Trial RandOmizing Heparin (ASTROH)-trial are highly anticipated (4). An additional residual dysfunctional coagulation in the subacute phase could prove protective against DCI by mitigating microthrombosis formation. In our comparison to matched controls, the rate of DCI was similar in patients on prior anticoagulant treatment. The subgroups of anticoagulated patients proved too small to focus analyses on DCI here. But, as our understanding of dysfunctional coagulation microvascular hypercoagulability improves also the potentially beneficial side-effects of iatrogenic anticoagulation may become clearer.”

  1. The finding of DOAC/VKA use not being an independent risk factor for poor outcome is interesting but I would have hoped for a little more development in the discussion of the findings in your tables.

We purposefully wanted the discussion to focus on the issue of anticoagulant’s safety in patients carrying unruptured aneurysms and hereby we limited the discussion of numerical results from our series. This because we believe this is the main area to which this analysis can contribute.

In the now revised text, we have highlighted the differences in ventilation and sepsis results in the discussion along an elaboration of the role coagulation plays in DCI pathophysiology. In response to Reviewer 2’s comments we discussed the possibility of differences in emergency services response time and risk of sudden death in SAH. We hope the reviewer understands and accepts this focus of ours.

Reviewer 2 Report

In this two-center observational study, the authors assessed the bleeding severity and outcome after SAH in patients on VKA and DOAC and reached several conclusions. These include that iatrogenic coagulopathy is not associated with severe SAH, that anticoagulant use is not associated with increased risk of poor outcomes, and that the rate of poor outcomes was similar between patients on VKA or DOAC.

While the methodology used in the study was sound, and the results significantly add to current medical knowledge, these findings are not easily generalizable considering the retrospective nature of the study. The authors equally note this in their limitations, as well as the presence of non-correctable confounders that may bias the results.

Author Response

In this two-center observational study, the authors assessed the bleeding severity and outcome after SAH in patients on VKA and DOAC and reached several conclusions. These include that iatrogenic coagulopathy is not associated with severe SAH, that anticoagulant use is not associated with increased risk of poor outcomes, and that the rate of poor outcomes was similar between patients on VKA or DOAC.

While the methodology used in the study was sound, and the results significantly add to current medical knowledge, these findings are not easily generalizable considering the retrospective nature of the study. The authors equally note this in their limitations, as well as the presence of non-correctable confounders that may bias the results.

We fully agree with Reviewer # 2’s comments and we have tried to phrase our conclusions as carefully as possible. The statement “in hospitalized patients” was applied consistently to point out the difference between patients suffering DCI-induced sudden death. We believe that, as our sample was drawn from an SAH population of almost 1000 patients, including more centers would only marginally have increased case numbers. The true shortcomings of this analysis will possibly only be addressed in either the long-term observation of cause of death in anticoagulant registries or population wide autopsy studies in the population at risk (on anticoagulants). Both of which have feasibility issues.

We have added a section to the discussion regarding Ringel et al.’s 1997 results of worse initial clinical outcome in anticoagulated SAH patients. This to introduce the effect of door-to-door time in emergency care as a possible limitations.

“Nonetheless, if rupture occurs, thrombus formation at the rupture site may take more time, presumably leading to more severe bleeding. Our series demonstrates how hospitalized patients on anticoagulants presenting with SAH are not more severely affected than patients without anticoagulant drugs. The opposite effect was demonstrated in a Dutch series of 15 SAH patients on anticoagulant of which 14 were either dead or highly dependent after 3 months follow up (5). The author’s explanation for this effect is the worse clinical condition patients are in from the outset. It is possible that a faster emergency service’s response time in densely populated areas (such as the Netherlands) could increase the number of patients reaching medical services alive, but in worse clinical state.”
